# Associations of serum DNA methylation levels of chemokine signaling pathway genes with mild cognitive impairment (MCI) and Alzheimer's disease (AD)

**Ting Zou**[1], **Xiaohui Zhou**[1]\*, **Qinwen Wang**[2], **Yongjie Zhao**[1], **Meisheng Zhu**[1], **Lei Zhang**[1], **Wei Chen**[1], **Pari Abuliz**[1], **Haijun Miao**[1], **Keyimu Kabinur**[1], **Kader Alimu**[1]

**1** Department of Geriatrics, The First Affiliated Hospital of Xinjiang Medical University, Urumqi, Xinjiang Province, China, **2** Ningbo Key Lab of Behavior Neuroscience, Zhejiang Provincial Key Laboratory of Pathophysiology, School of Medicine, Ningbo University, Ningbo, Zhejiang Province, China

\* zhouxiaohui858@sina.com

## Abstract

### Objective

To investigate the associations of serum DNA methylation levels of chemokine signaling pathway genes with Alzheimer's disease (AD) and mild cognitive impairment (MCI) in elderly people in Xinjiang, China, and to screen out genes whose DNA methylation could distinguish AD and MCI.

### Materials and methods

37 AD, 40 MCI and 80 controls were included in the present study. DNA methylation assay was done using quantitative methylation-specific polymerase chain reaction (qMSP). Genotyping was done using Sanger sequencing.

### Results

DNA methylation levels of *ADCY2*, *MAP2K1* and *AKT1* were significantly different among AD, MCI and controls. In the comparisons of each two groups, *AKT1* and *MAP2K1*'s methylation was both significantly different between AD and MCI ($p < 0.05$), whereas *MAP2K1*'s methylation was also significantly different between MCI and controls. Therefore, *AKT1*'s methylation was considered as the candidate serum marker to distinguish AD from MCI, and its association with AD was independent of *APOE* ε4 allele ($p < 0.05$). *AKT1* hypermethylation was an independent risk factor for AD and *MAP2K1* hypomethylation was an independent risk factor for MCI in logistic regression analysis ($p < 0.05$).

### Conclusion

This study found that the serum of *AKT1* hypermethylation is related to AD independently of *APOE* ε4, which was differentially expressed in the Entorhinal Cortex of the brain and was an independent risk factor for AD. It could be used as one of the candidate serum markers

**Data Availability Statement:** All relevant data are within the paper and its Supporting information files.

**Funding:** This research was supported by the grants from the National Natural Science Foundation of China (No. 81360064), which had role in study design, data collection. the Key project of Xinjiang Natural Science Foundation (No. 2022D01D63), which had role in decision to publish. National Key research and development program (2016YFC1305900), which were in charge of preparation of the manuscript in the study.

**Competing interests:** The authors have declared that no competing interests exist.

to distinguish AD and MCI. Serum of *MAP2K1* hypomethylation is an independent risk factor for MCI.

## Introduction

The incidence of aging-related diseases has increased significantly with the rapid progression of aging in society, which has increased the prevalence of neurological degenerative diseases rapidly, such as MCI and AD. MCI is a transitional stage of healthy aging to AD, which extent of cognitive decline and the clinical prophase has not reached the severity of AD yet [1]. 10% to 20% of MCI patients with clinical manifestations of memory impairment progress to AD each year [2, 3]. Since no efficient serum markers have been identified, cognitive function-related scales are usually used to distinguish AD and MCI.

DNA methylation is a major component of epigenetics, which is influenced by environmental factors leading to progression of diseases and providing new directions on pathogenesis and diagnosis of diseases. It could be the most promising blood marker for AD diagnosis in the future, and more effort should be devoted to the study of post-cellular DNA methylation [4]. Several studies have shown that the pathogenesis of AD and MCI is influenced by DNA methylation [5–9], but there are still few studies on DNA methylation of the same gene for comparison between AD and MCI.

Neuroimmune inflammation plays an important role in the pathogenesis of AD [10], and it has been found that it might be developed during the MCI phase [11, 12]. Chemokine signaling pathway is simultaneously conducted in astrocytes and microglia, which are the main sites of neuroinflammation responses. Therefore, it is an important correlated pathway in the mechanism of neuroimmune inflammation in AD [13–15].

At present, most research is on TREM2-DAP12 and CX3CL1-CX3CR1 axis in chemokine signaling pathway, which play an important role in neurodegenerative diseases and can regulate cognitive function and synaptic plasticity, especially in the hippocampus [16]. *CXCR5* gene of the CX3CL1-CX3CR1 axis has been found related to cognitive impairment [17], but its relationship with MCI or AD is still unclear, and there are few studies on DNA methylation differences of other genes in this pathway in AD and MCI.

Therefore, we used keywords to screen all the genes in the chemokine signaling pathway in the KEGG PATHWAY Database, and further selected the genes that had not been studied in DNA methylation through literature review and checked whether their CpG islands had methylation research value. Then seven genes (*CXCL5*, *ADCY2*, *HCK*, *MAP2K1*, *AKT1*, *WASL*, *RAP1B*) were screened out. Afterwards, in order to explore the associations of serum DNA methylation levels of the seven chemokine signaling pathway genes with AD and MCI, and further to screen out the genes which could distinguish AD from MCI, we investigated the associations of the serum DNA methylation levels in the promoter regions of the 7 genes with AD and MCI in Xinjiang, China. The genes whose DNA methylation levels were significantly different in AD were screened out and their expressions in the different brain regions in AD were verified by the AlzData database.

## Materials and methods

### Subjects

A total of 157 subjects were selected for the study, including 37 in the AD group, 40 in the MCI group and 80 in the control group. All of them were from a community-based

**Table 1. The baseline clinical data of the included subjects.**

| Characteristics | AD (n = 37) | MCI (n = 40) | Controls (n = 80) | p |
|---|---|---|---|---|
| Male / Female | 20 / 17 | 20 / 20 | 40 / 40 | 0.911 |
| Hypertension (Yes / No) | 22 / 15 | 19 / 21 | 49 / 31 | 0.341 |
| Diabetes (Yes / No) | 16 / 21 | 13 / 27 | 44 / 36 | 0.060 |
| Age (years) | 77 (74, 82) | 74.19 ± 8.03 | 75 (71.50, 81.00) | 0.128 |
| FBG (mmol / L) | 5.32 (5.06, 6.28) | 5.56 (5.05, 6.14) | 5.48 (4.90, 6.32) | 0.772 |
| TG (mmol / L) | 1.35 (0.93, 2.04) | 1.23 (0.98, 1.42) | 1.14 (0.91, 1.77) | 0.489 |
| TC (mmol / L) | 4.46 ± 1.28 | 4.70 ± 1.15 | 4.40 ± 1.08 | 0.416 |
| HDL (mmol / L) | 1.44 ± 0.40 | 1.62 ± 0.52 | 1.54 (1.30, 1.82) | 0.095 |
| LDL (mmol / L) | 3.03 ± 0.91 | 3.12 ± 0.84 | 2.90 ± 0.91 | 0.444 |
| MMSE | 19 (14,21) | 22 (20,24) | 27 (25,28) | **<0.001** |
| MoCA | 11 (5,15) | 18 (15, 20) | 23 (21, 25) | **<0.001** |

epidemiological survey of cognitive impairment elderly people aged ≥60 years in Urumqi, Xinjiang from 2017 to 2018. The general conditions and clinical data of the three groups were shown in Table 1. Ethics Committee of the First Affiliated Hospital of Xinjiang Medical University examined the project and found that it met the ethical requirements and approved the declaration. All the enrolled subjects had signed written informed consent forms.

All participants received neuropsychological tests to assess the level of cognitive. Neuropsychological tests in Chinese version included: the Mini-Mental State Examination (MMSE), the Montreal Cognitive Assessment Form (MoCA), Activities of Daily Living (ADLs) Scale, the overall Deterioration Scale (GDS), the Clinical Dementia Rating (CDR), and Kazakhstan Kinski ischemic Score (HIS) screening. Diagnosis criteria: A clinical diagnosis of AD or MCI was established according to the criteria of the Diagnostic and Statistical Manual-IV(DSM-IV;) [18]. Exclusion criteria for the current study were: (1) to exclude those with mental illness; (2) to exclude the brain dysfunction can cause neurological diseases such as cerebral hemorrhage, cerebral infarction, Parkinson's disease, intracranial tumors; (3) to exclude depression; (4) to exclude patients with severe cardiopulmonary liver and kidney dysfunction, severe infectious diseases, severe endocrine disease patients and toxic encephalopathy patients.

## Collection of blood samples and clinical data

Two venous blood samples were taken in the morning on an empty stomach from 157 subjects. One was used for the detection of general biochemical indicators, and the other whole blood was anticoagulated and stored in the refrigerator at -80°C for subsequent genomic DNA extraction. Clinical information was also collected from all subjects.

## Screening genes

All genes of the chemokine signaling pathway were screened out from the KEGG PATHWAY Database website. In PubMed, the keyword "Alzheimer's disease and gene name methylation" was used to search for all genes in the chemokine signaling pathway, and genes' DNA methylation that had not been studied in previous studies were retained. All retained genes were checked CpG islands in UCSC. The genes with CpG islands and a single band in the gel running experiment were selected through the gel running experiment. Seven genes which were *CXCL5, ADCY2, HCK, MAP2K1, AKT1, WASL* and *RAP1B* in the chemokine signaling pathway were finally selected.

## DNA preparation, methylation assay and genotyping

All the subjects' fasting venous blood was extracted in the morning, added with EDTA anticoagulant and stored in a refrigerator at -80°C. We extracted DNA of blood samples using a blood genomic DNA extraction kit (Omega Bio-tek, Inc. USA). The concentration and purity of the extracted DNA were measured with an ND1000 ultra-micro UV spectrophotometer (Nanodrop1000, Wilmington, USA).

The structure and CpG island (CGI) near by the promoter region of the seven genes were searched by PubMed and UCSC databases, and the corresponding DNA sequences were obtained to design gene primer sequences (S1 Table). Bisulfite transformation method was used to transform genomic DNA (EZ DNA Methylation-Gold™ kit) for accurate and rapid bisulfite methyl modification of DNA. Quantitative methylation-specific polymerase chain reaction (qMSP) was used to detect the level of DNA methylation (Roche LightCycler®480 instrument and LightCycler®480 SYBR Green I Master Mix kit). Genotyping of *APOE* rs7412 rs429358 was performed using Polymerase chain reaction (PCR) and Sanger sequencing. Annealing temperatures of qMSP and PCR were shown in S1 Table.

## Statistical analysis

SPSS 26.0 (SPSS, Inc., Chicago, IL, USA) software and R software (version 4.1.3) were used for statistical analysis, and $p < 0.05$ was considered statistically significant. Continuous variables are represented by X±S or M(IQR) depending on whether they are normally distributed or not. Two independent sample t-tests are used to compare normally distributed variables or continuous variables with data converted into a normal distribution. The Wilcox test was used for data that did not conform to normal distribution after conversion, and the chi-square test was used for comparison between classified variables. Spearman's rank correlation test was used to analyze the association between gene methylation and subjects' serum biochemical indicators.

## Validation of expressions of AD-related chemokine signaling pathway genes in different brain regions

The expressions of AD-related chemokine signaling pathway genes in different brain regions between AD and normal brain tissues were validated by AlzData (www.alzdata.org). AlzData is a useful database that provides a large number of human brain gene expression profiling [19, 20].

## Results

As shown in Table 1, there was no significant difference in gender, age, serum total cholesterol (TC), triglyceride (TG), high-density lipoprotein cholesterol (HDL-C), low-density lipoprotein cholesterol (LDL-C), hypertension and type 2 diabetes (T2DM) among AD group, MCI group and control group ($p>0.05$).

We compared the seven genes' DNA methylation levels among three groups first. DNA methylation levels of *ADCY2*, *MAP2K1*, and *AKT1* were significantly different among three groups (Table 2, $p < 0.05$); Breakdown analysis by gender showed that DNA methylation levels of *ADCY2*, *MAP2K1*, and *AKT1* were significantly different in males, and DNA methylation levels of *MAP2K1* and *AKT1* were significantly different in females (Table 2, $p < 0.05$). Further subgroup analysis by carrying *APOE* ε4 allele showed that *AKT1* methylation was significantly different in the non-*APOE* ε4 group (Table 2, $p < 0.05$).

**Table 2. Comparisons of seven genes' DNA methylation levels among AD group, MCI group and control group.**

| Genes | Methylation levels | | | p |
|---|---|---|---|---|
| | AD | MCI | Control | |
| **Total** | | | | |
| CXCL5 | 0.061(0.033, 0.108) | 0.054(0.011, 0.094) | 0.057(0.034, 0.085) | 0.568 |
| ADCY2 | 0.018(0.010, 0.032) | 0.017(0.007, 0.044) | 0.026(0.013, 5.697) | **0.013** |
| HCK | 0.007(0.002, 0.016) | 0.011(0.005, 0.023) | 0.013(0.007, 0.020) | 0.063 |
| MAP2K1 | 1.369(1.063, 1.822) | 0.328(0.192, 0.590) | 0.724(0.462, 1.394) | **<0.001** |
| AKT1 | 1.453(1.051,1.727) | 0.616(0.334, 0.938) | 0.765(0.380, 1.104) | **<0.001** |
| WASL | 5.276(1.808, 12.215) | 4.064(0.820, 9.223) | 7.140(4.741, 10.380) | 0.06 |
| RAP1B | 8.854(4.225, 12.100) | 6.047(2.345, 8.893) | 7.376(4.116, 11.770) | 0.145 |
| **Males** | | | | |
| CXCL5 | 0.057(0.031, 0.080) | 0.052(0.012, 0.094) | 0.050(0.036, 0.094) | 0.820 |
| ADCY2 | 0.016(0.010, 0.026) | 0.028(0.012, 0.050) | 0.036(0.016, 7.298) | **0.022** |
| HCK | 0.011(0.002, 0.022) | 0.016(0.005, 0.032) | 0.012(0.006, 0.021) | 0.369 |
| MAP2K1 | 1.345(1.103, 1.820) | 0.313(0.229, 0.574) | 1.192(0.559, 1.486) | **<0.001** |
| AKT1 | 1.215(1.036, 1.578) | 0.740(0.380, 0.903) | 0.881(0.500, 1.178) | **<0.001** |
| WASL | 3.084 (1.245, 10.529) | 3.716(0.688, 7.077) | 7.545(4.662, 10.260) | 0.067 |
| RAP1B | 10.208(4.191, 11.848) | 6.564(0.919, 9.735) | 7.611(4.321, 11.570) | 0.200 |
| **Females** | | | | |
| CXCL5 | 0.086(0.042, 0.132) | 0.057(0.006, 0.148) | 0.063(0.031, 0.084) | 0.298 |
| ADCY2 | 0.028(0.008, 0.036) | 0.013(0.003, 0.029) | 0.022(0.011, 1.552) | 0.153 |
| HCK | 0.007 ± 0.006 | 0.010(0.005, 0.018) | 0.013(0.008, 0.018) | 0.056 |
| MAP2K1 | 1.561 ± 0.889 | 0.353(0.082, 0.737) | 0.504(0.378, 1.276) | **<0.001** |
| AKT1 | 1.518(1.014, 1.826) | 0.664 ± 0.462 | 0.765(0.384, 1.109) | **<0.001** |
| WASL | 5.815(3.133, 14.985) | 4.945(1.155, 10.970) | 6.745(4.749, 10.778) | 0.663 |
| RAP1B | 6.777(3.905, 15.355) | 5.420(4.016, 6.971) | 6.956(2.684, 11.838) | 0.665 |
| **APOE ε4+** | | | | |
| CXCL5 | 0.059 ± 0.046 | 0.091(0.026, 0.151) | 0.073(0.048, 0.102) | 0.434 |
| ADCY2 | 0.006(0.000, 0.019) | 0.030 ± 0.026 | 0.020(0.015, 8.418) | 0.061 |
| HCK | 0.002(0.001, 0.023) | 0.011(0.007, 0.064) | 0.013(0.008, 0.045) | 0.276 |
| MAP2K1 | 2.033 ± 1.572 | 0.401 ± 0.280 | 0.699(0.332, 1.336) | **0.003** |
| AKT1 | 1.508 ± 1.087 | 0.763 ± 0.451 | 0.648(0.376, 0.934) | 0.071 |
| WASL | 3.088(1.102, 13.854) | 5.309 ± 4.379 | 10.514 ± 6.938 | 0.087 |
| RAP1B | 4.900(2.381, 16.105) | 6.371(3.509, 14.050) | 7.073(3.541, 11.045) | 0.998 |
| **APOE ε4-** | | | | |
| CXCL5 | 0.064(0.033, 0.108) | 0.049(0.010, 0.091) | 0.050(0.032, 0.084) | 0.168 |
| ADCY2 | 0.023(0.013, 0.037) | 0.016(0.010, 0.040) | 0.029(0.012, 4.900) | 0.062 |
| HCK | 0.008(0.003, 0.015) | 0.011(0.005, 0.020) | 0.013 (0.006, 0.018) | 0.105 |
| MAP2K1 | 1.344(1.042, 1.818) | 0.328(0.197, 0.590) | 0.799(0.471, 1.409) | **<0.001** |
| AKT1 | 1.463(1.068, 1.726) | 0.622(0.327, 0.913) | 0.770(0.392, 1.128) | **<0.001** |
| WASL | 5.311(2.095, 12.438) | 3.871(0.827, 8.914) | 6.579(4.662, 9.306) | 0.26 |
| RAP1B | 8.964(5.143, 11.848) | 5.552 ± 3.902 | 7.473(4.315, 11.990) | 0.057 |

In the comparisons of each two groups, it showed that *ADCY2*, *MAP2K1* and *AKT1*'s methylation levels were significantly different between AD and control (Fig 1, $p < 0.05$). We found that *ADCY2* methylation was significantly decreased in AD group, while *MAP2K1* methylation and *AKT1* methylation were significantly increased in AD group. Further subgroup analysis by carrying *APOE* ε4 allele showed that DNA methylation levels of *MAP2K1* and *AKT1* were

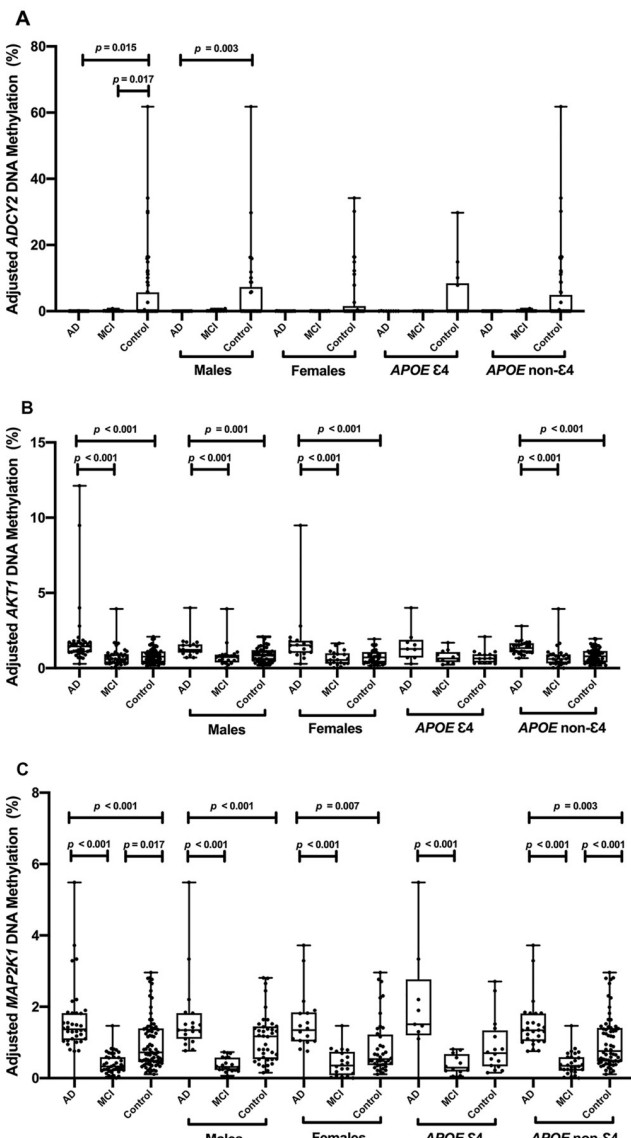

**Fig 1. Comparisons of *ADCY2*, *AKT1* and *MAP2K1's* DNA methylation levels between any two groups.** ε4 refers to subject carrying at least one *APOE* ε4 allele; non-ε4 refers to subject carrying no *APOE* ε4 allele.

both associated with AD independent of *APOE* ε4 (Fig 1B and 1C, $p < 0.05$). *ADCY2* and *MAP2K1*'s methylation levels were significantly different between MCI and control, and both of them were decreased in MCI group (Fig 1A and 1C, $p < 0.05$). *AKT1* and *MAP2K1*'s methylation levels were significantly different between AD and MCI, and both of them were increased in AD group (Fig 1B and 1C, $p < 0.05$). Therefore, *MAP2K1* methylation was both significantly different in AD and MCI (Fig 1C, $p < 0.05$), while *AKT1* methylation was only significantly different in AD (Fig 1B, $p < 0.05$). That meant *AKT1* methylation was helpful to distinguish AD from MCI, and its association with AD was also independent of *APOE* ε4 allele.

The three AD-related genes' expressions were validated in the normalized brain gene expression profile of AlzData (www.alzdata.org). It showed that *ADCY2* didn't have

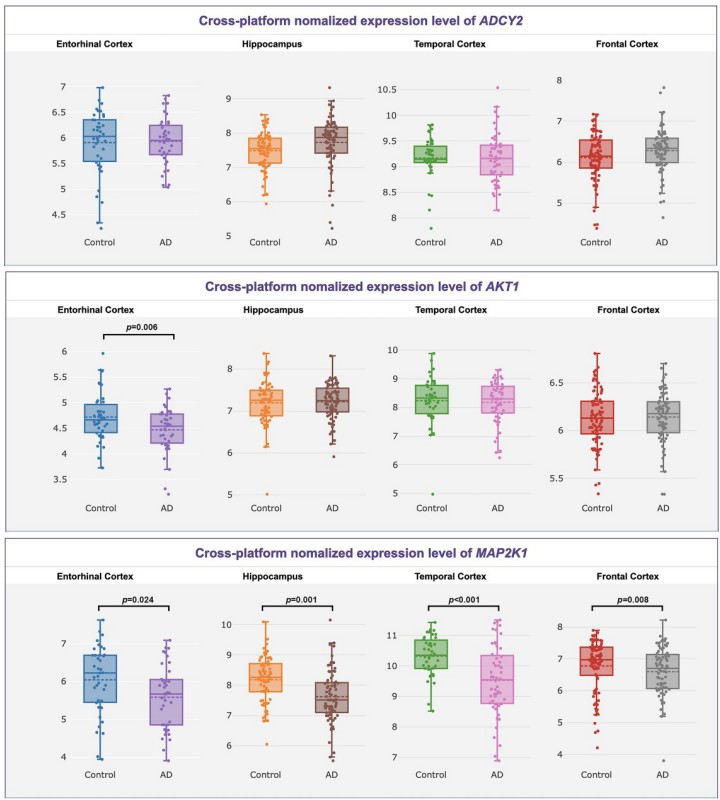

**Fig 2. Validation of the *ADCY2*, *MAP2K1*'s and *AKT1* brain expression in AlzData (http://www.alzdata.org/index.html).**

significantly different expressions in the brain between AD and normal people (Fig 2, $p > 0.05$), while *AKT1* had significantly different expressions in Entorhinal Cortex and *MAP2K1* had significantly different expressions in Entorhinal Cortex, Temporal Cortex, Hippocampus and Frontal Cortex in AD (Fig 2, $p < 0.05$).

The AD-related and MCI-related genes were included in the logistic regression equation analysis, and it showed that *AKT1* hypermethylation was an independent risk factor for AD, and *MAP2K1* hypomethylation was an independent risk factor for MCI (Table 3, $p < 0.05$). Since DNA methylation is affected by environmental factors, in order to explore whether there are any related biochemical indicators affect their DNA methylation levels, we further analyzed

**Table 3. The logistics regression of *ADCY2*, *MAP2K1* and *AKT1*'s DNA methylation in AD and MCI respectively.**

|  |  | β | Std. Error | Wald | *p* | Exp(B) | 95% CI |
|---|---|---|---|---|---|---|---|
| AD | (Intercept) | -0.276 | 1.062 | 0.068 | 0.795 |  |  |
|  | *ADCY2* | -0.896 | 2.097 | 0.183 | 0.669 | 0.408 | 0.007–24.859 |
|  | *MAP2K1* | 0.081 | 0.332 | 0.059 | 0.808 | 1.084 | 0.565–2.081 |
|  | *AKT1* | 1.550 | 0.566 | 7.501 | **0.006** | 4.713 | 1.554–14.292 |
| MCI | (Intercept) | 1.080 | 1.144 | 0.892 | 0.345 |  |  |
|  | *ADCY2* | -1.810 | 1.505 | 1.446 | 0.229 | 0.164 | 0.009–3.128 |
|  | *MAP2K1* | -4.560 | 0.903 | 25.474 | **0.000** | 0.010 | 0.002–0.061 |
|  | *AKT1* | 0.994 | 0.619 | 2.579 | 0.108 | 2.703 | 0.803–9.098 |

correlations of *AKT1* and *MAP2K1*'s methylation levels and serum biochemical indexes in AD and MCI respectively. We found that *AKT1* hypermethylation was not associated with serum biochemical indicators in AD (S1 Fig, $p > 0.05$), and *MAP2K1* hypomethylation was negatively correlated with serum TG levels in the MCI female group (S2 Fig, $p = 0.0084$, r = -0.58).

## Discussion

Numerous studies have confirmed that neuroinflammation plays an important role in the pathogenesis of AD [21–24], and anti-neuroinflammation can reduce cognitive impairment in AD animal models [25] MCI is the preclinical stage of AD, and studies have found that the neuropathological changes of MCI partially overlap with those of AD, and neuroinflammatory lesions have gradually been confirmed to have formed in the MCI stage [11, 12]. This early inflammation promotes and exacerbates the production of Aβ and NFT, and further leads to neuronal toxicity and death [26–28], which could make progress from MCI to AD.

Chemokine signaling pathway plays an important role in neuroinflammation, which is involved in signal transduction in astrocytes and microglia, and then participates in or directly leads to the occurrence of neuroinflammation, which leads to cognitive decline. As mentioned above, most current studies focus on TREM2-DAP12 and CX3Cl1-CX3CR1 axes. It has been found that the functional changes of the CX3CL1/CX3CR1 in different pathological conditions may promote the activation of microglia and stimulate the release of inflammatory factors [29–31]. *TREM2* gene acts downstream of CD33 and is involved in the regulation of Aβ pathology and neurodegeneration associated with AD risk [32]. However, there are few studies on other genes of this pathway and fewer studies on DNA methylation associated with AD and MCI. The seven genes selected in this study are all from chemokine signaling pathways and are directly or indirectly involved in neuroinflammation. However, there are still few studies on these genes, and the mechanism through which they act on neuroinflammatory response and the relationship between them are still to be explored.

Our results showed that the serum DNA methylation levels of three genes (*ADCY2*, *MAP2K1* and *AKT1*) in the chemokine signaling pathway were correlated with AD, and expressions of *MAP2K1* and *AKT1* were both verified differently in the brain of AD, which could be used as serum candidate markers for AD. We found that the DNA methylation levels of *AKT1* can be considered as one of the candidate markers to distinguish AD from MCI. Since carrying the *APOE* ε4 allele is an important risk factor for AD, we also conducted a further stratified analysis according to carry *APOE* ε4 whether or not. The results showed that *AKT1* serum DNA hypermethylation was associated with AD independent of *APOE* ε4.

*AKT1* encodes a serine/threonine kinase that is a central node in signaling pathways that regulate cell survival. Insufficient activity of the kinase Akt1 can lead to neuronal death, and it is one of the important proteins involved in the neuroinflammatory response of Alzheimer's disease. Recent studies have shown that its neuroinflammation was regulated by microRNAs and long non-coding RNAs, and controlled by interacting networks with other proteins [33]. *AKT1* methylation has been found to promote AKT kinase activity, and its effect on the histone methyltransferase SETDB1 can cause tumorigenesis [34]. However, there have been no studies linking *AKT1* methylation with neuroinflammation or cognitive dysfunction. The present study firstly found that its serum DNA methylation levels were significantly associated with AD independent of *APOE* ε4 and were differentially expressed in the Entorhinal Cortex of the brain in AD patients, but the sample size of this study is moderate and it needs to be verified in a large sample size.

Another finding of present study was that *MAP2K1* methylation was significantly different both in AD and MCI, and it was differentially expressed in multiple regions of the brain in

AD, including Entorhinal Cortex, Temporal Cortex, Hippocampus and Frontal Cortex. Although it couldn't be used as a marker to distinguish AD from MCI, logistic regression showed that it was an independent risk factor for MCI, so it could be used as one of the serum candidate markers for MCI. The *MAP2K1* gene encodes MAP2K1 protein, one of the signaling proteins of mitogen-activated protein kinases (MAPKs) which regulated almost all stimulated cellular processes, including proliferation, differentiation, and stress responses [35, 36]. Dysregulation of these kinases participated in many pathological process, including neurological diseases [37], such as Huntington's disease, multiple sclerosis, ischemia and cerebral hypoxia. However, few studies have been conducted on its association with cognitive dysfunction. This study firstly found that its DNA methylation is significantly different in AD and MCI, which can be further verified in larger samples and functional experiments to clarify its role.

However, due to the impact of funds, sample storage and logistics during the COVID-19 pandemic, this study could not further determine the protein expression levels of *AKT1* and *MAP2K1* genes in blood, nor could it conduct a complete detection of chemokines in the chemokine signaling pathway in blood, so it could not fully explore their role and mechanism in the pathogenesis of AD or MCI, which were limitations of this study.

## Conclusions

The present study found that the serum of *AKT1* hypermethylation is related to AD independently of *APOE* ε4, which was differentially expressed in Entorhinal Cortex of the brain and was an independent risk factor for AD. It could be used as one of the candidate genes to distinguish AD and MCI. Serum of *MAP2K1* hypomethylation is an independent risk factor for MCI and serum TG levels have negative correlation with it.

## Supporting information

**S1 Fig. Correlation of *AKT1* DNA methylation and biochemical criterion in AD.**
(TIF)

**S2 Fig. Correlation of *MAP2K1* DNA methylation and biochemical criterion in MCI.**
(TIF)

**S1 Table. Primer sequences for qMSP and single nucleotide polymorphism analysis of *APOE*.**
(DOCX)

**S1 Data.**
(XLSX)

## Acknowledgments

We are grateful to staff who joined in the epidemiological survey in 2018 in Urumqi, Xinjiang province, China.

## Author Contributions

**Conceptualization:** Xiaohui Zhou.

**Data curation:** Ting Zou.

**Formal analysis:** Ting Zou.

**Funding acquisition:** Xiaohui Zhou, Qinwen Wang.

**Investigation:** Ting Zou, Yongjie Zhao, Meisheng Zhu, Lei Zhang, Wei Chen, Pari Abuliz, Haijun Miao, Keyimu Kabinur, Kader Alimu.

**Methodology:** Ting Zou, Yongjie Zhao.

**Resources:** Yongjie Zhao, Meisheng Zhu, Lei Zhang, Wei Chen, Pari Abuliz, Haijun Miao, Keyimu Kabinur, Kader Alimu.

**Supervision:** Xiaohui Zhou.

**Writing – original draft:** Ting Zou.

**Writing – review & editing:** Ting Zou.

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
