## [Decision Letter · Decision Letter 0]

7 Aug 2023

PONE-D-23-21380Associations of serum DNA methylation levels of chemokine signaling pathway genes with mild cognitive impairment (MCI) and Alzheimer's disease (AD)PLOS ONE

Dear Dr. Zou,

Thank you for submitting your manuscript to PLOS ONE. After careful consideration, we feel that it has merit but does not fully meet PLOS ONE’s publication criteria as it currently stands. Therefore, we invite you to submit a revised version of the manuscript that addresses the points raised during the review process.

This paper in its present form does not reach enough level for acceptance in the Journal.  Major revisions are needed according to the Reviewers' comments.  Respond to them appropriately.

We look forward to receiving your revised manuscript.

Kind regards,

Masaki Mogi

Academic Editor

PLOS ONE

Journal Requirements:

"This research was supported by the grants from the National Natural Science Foundation of China (No. 81360064), the Key project of Xinjiang Natural Science Foundation (No. 2022D01D63), National Key research and development program (2016YFC1305900)"

Reviewers' comments:

Reviewer's Responses to Questions

**Comments to the Author**

1. Is the manuscript technically sound, and do the data support the conclusions?

Reviewer #1: Partly

2. Has the statistical analysis been performed appropriately and rigorously? 

Reviewer #1: Yes

3. Have the authors made all data underlying the findings in their manuscript fully available?

Reviewer #1: Yes

4. Is the manuscript presented in an intelligible fashion and written in standard English?

Reviewer #1: Yes

5. Review Comments to the Author

Reviewer #1: In this manuscript entitled “Associations of serum DNA methylation levels of chemokine signaling pathway genes with mild cognitive impairment (MCI) and Alzheimer's disease (AD),” the authors investigated the change of DNA methylation levels in the serum of MCI and AD patients. The subject of this study is interesting; however, several concerns were raised as follows:

Major comments:

1. In the introduction, the authors did not summarize the previous research precisely. The authors should mention previous studies and describe the significance of the subject of this study clearly.

2. The authors determined the DNA methylation levels of the seven genes; however, their logic and bases were unclear. The authors should describe it clearly. In addition, did these protein levels change in the blood of MCI and AD patients? Were these changes related to the pathogenesis of these diseases? What’s the specificity?

3. The authors identified significant difference of ADCY2, MAP2K1 and AKT1’s DNA methylation levels. Were these levels related to the seriousness or score of MCI and AD patients? How about the duration of these diseases?

4. In the title of this manuscript, “chemokine signaling pathway” was described. Were chemokines changed in the MCI and AD patients of the present study? Was the change in ADCY2, MAP2K1, and AKT1’s DNA methylation levels related to these cytokines?

6. PLOS authors have the option to publish the peer review history of their article (what does this mean?). If published, this will include your full peer review and any attached files.

Reviewer #1: No

---

## [Author Response · Author response to Decision Letter 0]

6 Oct 2023

DearReviewers：

Thank you for your letter and for the comments concerning our manuscript entitled “Associations of serum DNA methylation levels of chemokine signaling pathway genes with mild cognitive impairment (MCI) and Alzheimer's disease (AD)”. (ID: PONE-D-23-21380). Those comments are all valuable and very helpful for revising and improving our paper, as well as the important guiding significance to our researches. We have studied comments carefully and have made corrections which we hope meet with approval. Revised portions are marked with highlights in the paper. The main corrections in the paper and the responds to the reviewer’s comments were written on the letter of Response to reviewers. We look forward to your attention.

---

## [Decision Letter · Decision Letter 1]

20 Nov 2023

Associations of serum DNA methylation levels of chemokine signaling pathway genes with mild cognitive impairment (MCI) and Alzheimer's disease (AD)

PONE-D-23-21380R1

Dear Dr. Zou,

We’re pleased to inform you that your manuscript has been judged scientifically suitable for publication and will be formally accepted for publication once it meets all outstanding technical requirements.

Kind regards,

Masaki Mogi

Academic Editor

PLOS ONE

Additional Editor Comments (optional):

Reviewers' comments:

Reviewer's Responses to Questions

**Comments to the Author**

1. If the authors have adequately addressed your comments raised in a previous round of review and you feel that this manuscript is now acceptable for publication, you may indicate that here to bypass the “Comments to the Author” section, enter your conflict of interest statement in the “Confidential to Editor” section, and submit your "Accept" recommendation.

Reviewer #1: All comments have been addressed

2. Is the manuscript technically sound, and do the data support the conclusions?

Reviewer #1: (No Response)

3. Has the statistical analysis been performed appropriately and rigorously? 

Reviewer #1: (No Response)

4. Have the authors made all data underlying the findings in their manuscript fully available?

Reviewer #1: (No Response)

5. Is the manuscript presented in an intelligible fashion and written in standard English?

Reviewer #1: (No Response)

6. Review Comments to the Author

Reviewer #1: I am fully satisfied with the response of the authors. Therefore, I would like to recommend the publication of this manuscript.

7. PLOS authors have the option to publish the peer review history of their article (what does this mean?). If published, this will include your full peer review and any attached files.

Reviewer #1: No

---

## [Editor Report · Acceptance letter]

23 Nov 2023

PONE-D-23-21380R1 

Associations of serum DNA methylation levels of chemokine signaling pathway genes with mild cognitive impairment (MCI) and Alzheimer's disease (AD) 

Dear Dr. Zou:

I'm pleased to inform you that your manuscript has been deemed suitable for publication in PLOS ONE. Congratulations! Your manuscript is now with our production department. 

Kind regards, 

on behalf of

Dr. Masaki Mogi 

Academic Editor

PLOS ONE